# Endobronchial Ultrasonography with a Guide Sheath Transbronchial Biopsy for Diagnosing Peripheral Pulmonary Lesions within or near Fibrotic Lesions in Patients with Interstitial Lung Disease

**DOI:** 10.3390/cancers13225751

**Published:** 2021-11-17

**Authors:** Takayasu Ito, Shotaro Okachi, Tomoki Kimura, Kensuke Kataoka, Yasuhiko Suzuki, Fumie Kinoshita, Keiko Wakahara, Naozumi Hashimoto, Yasuhiro Kondoh

**Affiliations:** 1Department of Respiratory Medicine and Allergy, Tosei General Hospital, Seto 489-8642, Japan; takaito9@med.nagoya-u.ac.jp (T.I.); tomoki_kimura@tosei.or.jp (T.K.); kataoka@tosei.or.jp (K.K.); kondoh@tosei.or.jp (Y.K.); 2Department of Respiratory Medicine, Nagoya University Graduate School of Medicine, Nagoya 466-8560, Japan; wakahara@med.nagoya-u.ac.jp (K.W.); hashinao@med.nagoya-u.ac.jp (N.H.); 3Department of Pathology, Tosei General Hospital, Seto 489-8642, Japan; ysuzuki@tosei.or.jp; 4Department of Advanced Medicine, Nagoya University Hospital, Nagoya 466-8560, Japan; f-kinoshita@med.nagoya-u.ac.jp

**Keywords:** bronchoscopy, endosonography, lung neoplasms, pulmonary fibrosis

## Abstract

**Simple Summary:**

Lung cancer often occurs around fibrotic lesions in patients with interstitial lung disease (ILD). In patients with ILD, several methods are available for diagnosing peripheral pulmonary lesions (PPLs), such as bronchoscopy with radial endobronchial ultrasound (R-EBUS), transthoracic needle biopsy, and surgical lung biopsy. As well as previous reports, in patients with ILD, bronchoscopy with R-EBUS might be an option as the primary procedure for diagnosing PPLs with fewer complications. However, the utility and safety of bronchoscopy with R-EBUS for PPLs in patients with ILD remain unknown. In this study, we assessed the efficacy and complications as the initial diagnostic procedure of bronchoscopy with R-EBUS according to the proximity of PPLs to fibrotic lesions. Our study might make a contribution to physicians who treat PPLs in patients with underlying ILD.

**Abstract:**

In patients with interstitial lung disease (ILD), the most frequent locations of lung cancer are within or near fibrotic lesions. However, the diagnostic yield for peripheral pulmonary lesions (PPLs) within or near fibrotic lesions using endobronchial ultrasonography with a guide sheath transbronchial biopsy (EBUS-GS TBB) may be unsatisfactory compared to that for PPLs distant from fibrotic lesions because of the difficulty in reaching the lesions. Our objectives were to evaluate the yield for PPLs using EBUS-GS TBB according to the proximity of PPLs to fibrotic lesions and to determine factors affecting the yield for PPLs. We retrospectively investigated 323 consecutive lesions using EBUS-GS TBB between 1 November 2014 and 31 December 2016. We identified PPLs with ILD in such lesions. PPLs with ILD were divided into PPLs within or near fibrotic lesions which met the criterion of PPLs, and of fibrotic lesions overlapping each other (PPLs-FL) and those distant from fibrotic lesions, which met the criterion of PPLs and the area of fibrotic lesion not overlapping each other (PPLs-NFL). Of the 323 lesions, 55 were included (31 PPLs-FL and 24 PPLs-NFL). The diagnostic yield for PPLs-FL was significantly lower than for PPLs-NFL (45.2% vs. 83.3%, *p* = 0.004). Multivariate analysis revealed that PPLs-NFL (odds ratio (OR) = 7.509) and a probe position within the lesion (OR = 4.172) were significant factors affecting diagnostic yield. Lesion’s positional relation to fibrotic lesions and the probe position were important factors affecting the successful diagnosis via EBUS-GS TBB in these patients.

## 1. Introduction

Lung cancer is the leading cause of cancer-related mortality worldwide, and it often occurs in patients with interstitial lung disease (ILD), including idiopathic pulmonary fibrosis (IPF) [1]. The most frequent locations of lung cancer in patients with ILD are the lower lobe, the peripheral lung, and within or near the honeycomb formation [2]. However, all lesions occurring in patients with ILD are not always malignant. Therefore, it is important to obtain a correct histological diagnosis by sampling the lesions [3]. Furthermore, in determining the treatment policy in patients with lung cancer co-existing with IPF, surgery can exacerbate IPF, while radiation or chemotherapy are also risk factors for the development of pulmonary fibrosis [4,5,6]. Therefore, correct histological diagnosis is more important to present appropriate treatment policy to patients with ILD.

In patients with ILD, several methods are available for diagnosing PPLs, such as bronchoscopy, transthoracic needle biopsy (TTNB), and surgical lung biopsy (SLB) [7]. However, the optimal technique for the diagnosis of PPLs in patients with ILD remains unknown [8,9]. Recently, the diagnostic yield for PPLs has been improved by bronchoscopy with radial endobronchial ultrasound (R-EBUS) and a navigation system compared to conventional bronchoscopy [7,10,11,12,13,14]. Bronchoscopy with R-EBUS and a navigation system might be another option as the primary procedure for diagnosing PPLs with fewer complications in patients with ILD compared to TTNB and SLB [7,12]. However, the utility and safety of endobronchial ultrasonography with a guide sheath transbronchial biopsy (EBUS-GS TBB) for PPLs in patients with ILD remain unknown.

We hypothesised that in patients with ILD, diagnostic yield for PPLs located within or near fibrotic lesions using EBUS-GS TBB might be unsatisfactory compared to that for PPLs located distant from fibrotic lesions because of the difficulty in reaching the lesions in relation to anatomical changes and the difficulty in finding lesions by fluoroscopy. Hence, we evaluated the diagnostic yield and complications of EBUS-GS TBB between PPLs located within or near fibrotic lesions and those distant from fibrotic lesions, along with factors affecting the diagnostic yield for PPLs.

## 2. Materials and Methods

### 2.1. Patient Enrolment

We introduced radial EBUS in our department on 1 November 2014. Therefore, we performed a retrospective analysis of consecutive patients who underwent EBUS-GS TBB for PPLs at Tosei General Hospital between 1 November 2014 and 31 December 2016. We included PPLs that were defined as lesions surrounded by normal lung parenchyma or interstitial lung area and not visible by bronchoscopy. Patients with an unknown final diagnosis, as described below, were excluded. In patients subjected to EBUS-GS for PPLs, we selected patients with ILD. The diagnosis of ILD was based on previously established criteria and expert evaluation and integration of data obtained from the patient’s history, physical examination, HRCT, and pulmonary function tests [15,16,17]. In our study, PPLs in patients with ILD were divided into two groups as follows: a) PPLs within or near fibrotic lesions (PPLs and the area of fibrotic lesions overlapped each other) and b) PPLs distant from fibrotic lesions (PPLs and the area of fibrotic lesions did not overlap each other) (Figure 1) [18]. Written informed consent was obtained from all patients prior to undergoing bronchoscopy. This study was approved by the Tosei General Hospital Institutional Review Board (IRB#960).

### 2.2. EBUS-GS TBB Procedure

Pulmonary function test (PFT) (spirometry and diffusing capacity of the lung for carbon monoxide (DL_CO_)) was performed within 4 weeks before bronchoscopy in the majority of cases. Before the procedure, all patients were locally anaesthetized with a 2% lidocaine spray, and an intravenous bolus of midazolam and fentanyl was administered. Then, a thin bronchoscope (BF-P260F; Olympus, Tokyo, Japan) with a guide sheath (K-201; Olympus; external diameter, 1.95 mm) was used for the 1.4-mm probe. After the probe was inserted and the R-EBUS image was confirmed, the probe was withdrawn, and a transbronchial forceps biopsy (FB-233D; Olympus) was repeated until an adequate number of specimens had been sampled. We classified the EBUS probe positions into three groups as follows: (a) within, when the probe was located inside the PPL; (b) adjacent to, when the probe was located at the periphery of the PPL; and (c) invisible when the probe was located away from the PPL. We used a virtual bronchoscopic navigation system (Bf-NAVI; Cybernet Systems, Tokyo, Japan) from the helical CT data with a slice width of 0.5 mm in most cases. In our study, we did not use rapid on-site evaluation during EBUS-GS TBB.

### 2.3. Variables

The following clinical information was collected from all patients who underwent the procedure: age, sex, smoking history, classification of ILD, lesion size, lesion lobe, lesion location, lesion structure, bronchus sign on CT, PFT, visibility on chest X-ray, EBUS image, number of biopsies, bronchoscopic diagnosis, and final diagnosis. ILD was classified into two groups: IPF or non-IPF. The lesion location from the hilum was classified into two groups: inner, for lesions in the inner- and middle-third ellipses; and outer, for lesions in the outer-third ellipse [19]. The structure of the lesion was classified into the three groups as follows: solid, part-solid, or pure ground-glass [20]. The bronchus sign represented detection of a bronchus directly leading to the lesion on CT [21]. We confirmed the final pathological diagnosis and microbiological analysis obtained by biopsy via bronchoscopy, TTNB, or SLB, and clinical follow-up. Successful diagnosis obtained using bronchoscopy was defined as malignant lesions based on histopathology. In contrast, a failed diagnosis was defined as a case where the sample was inadequate (e.g., peripheral lung tissue or peribronchial tissue). When the collected specimens showed specific benign findings (e.g., granuloma) and the subsequent clinical course was assessed to be radiologically decreased in size or become stable during the follow-up period of more than one year after the procedure, bronchoscopy was counted as diagnostic. Conversely, for PPLs that had grown within one year after the procedure, a definite diagnosis was obtained by additional diagnostic modalities (i.e., TTNB or SLB). In these cases, if the initial diagnosis via bronchoscopy was inconsistent with the final diagnosis at re-examination, bronchoscopy was considered non-diagnostic. Benign lesions, which could not be pathologically or microbiologically diagnosed, were evaluated by confirming radiologic size stability during the follow-up period for at least one year after bronchoscopy. If the lesions of part-solid or pure ground-glass structures were undiagnosed by bronchoscopy, follow-up was done using CT according to Fleischner Society guidelines, and definite diagnosis and appropriate therapy were obtained by operation [22].

### 2.4. Statistical Analysis

Data are presented as medians and ranges. Mann–Whitney U tests and Pearson chi-squared tests were used to analyse continuous and categorical variables, respectively. Multivariate logistic regression analyses were performed to investigate significant predictors of positive results with EBUS-GS TBB. The variables analysed in relation to diagnosis using EBUS-GS TBB were as follows: (a) the positional relation of PPLs to fibrotic lesions (PPLs within or near fibrotic lesions, or those distant from fibrotic lesions), (b) largest diameter (≤20 mm or >20 mm), (c) lesion location (inner or outer), (d) bronchus sign (positive or negative), and e) EBUS image (within, others). In our study, we selected the size of 20 mm as a cut-off value in the multivariate logistic regression analysis according to the previous meta-analysis [7]. The statistical significance was set at *p* < 0.05, and all reported *p* values were two-sided. All analyses were performed using SPSS Statistics (version 28; IBM, Armonk, NY, USA).

## 3. Results

### 3.1. Patient Characteristics

Among the 323 PPLs evaluated by EBUS-GS TBB, 61 lesions were identified in patients with ILD. After excluding two endobronchial lesions and four lesions because of uncertain diagnosis, 55 lesions were finally included in the analyses. Among these, 31 PPLs were within or near fibrotic lesions, and 24 lesions were distant from fibrotic lesions (Figure 2). The characteristics of the patients and their lesions in the two groups are presented in Table 1. There was a significant difference in age, smoking history, classification of ILD, lesion location between the groups (Table 1).

### 3.2. Diagnostic Yields by Probe Location and Disease Type between PPLs within or near Fibrotic Lesions and Those Distant from Fibrotic Lesions

The diagnostic yield for PPLs within or near fibrotic lesions was significantly lower than that for PPLs distant from fibrotic lesions (45.2% vs. 83.3%, *p* = 0.004). The location of the probe within the lesion was no more frequent between PPLs within or near fibrotic lesions and those distant from fibrotic lesions (51.6% vs. 50%, respectively). When the probe was located within the lesion, the diagnostic yield for PPLs within or near fibrotic lesions was significantly lower than that for those distant from fibrotic lesions (56.3% vs. 100%, *p* = 0.023). Furthermore, the diagnostic yield for malignant lesions was significantly lower in PPLs within or near fibrotic lesions than in those distant from fibrotic lesions (41.7% vs. 100%, *p* < 0.001) (Table 2). In the 17 lesions undiagnosed by EBUS-GS, TTNB performed as further additional procedures gave a final diagnosis of 10 lesions consisting of one lesion of adenocarcinoma, two lesions of squamous cell carcinoma, four lesions of small cell lung carcinoma, and three lesions of non-small cell lung carcinoma, while SLB gave a diagnosis of four lesions consisting of three lesions of adenocarcinoma, and one lesion of adenosquamous cell carcinoma. On the other hand, three of the 17 lesions were evaluated via follow-ups with CT, and all three lesions regressed spontaneously six months after bronchoscopy; these lesions were diagnosed as inflammatory lesions. In these additional diagnostic procedures, the diagnostic accuracy of TTNB and SLB was 10/10 (100%) and 4/4 (100%), respectively, whereas the rate of complication of TTNB and SLB was 3/10 (30%) and 0/4 (0%), respectively. Pneumothorax not requiring drainage was identified as the complication encountered when TTNB was performed as an additional procedure.

### 3.3. Factors Related to Successful Diagnosis of PPLs by EBUS-GS TBB in Patients with ILD

In multivariate analyses, the positional relation of PPLs to fibrotic lesions and the probe position on EBUS image were significant predictors of successful diagnosis by EBUS-GS TBB (Table 3).

### 3.4. Complications

Pneumothorax occurred in seven patients (12.7%). Of these, pneumothorax occurred in five patients (16.1%) with PPLs within or near fibrotic lesions, with three (9.7%) patients requiring thoracic drainage. Overall, there were two patients (8.3%) with PPLs distant from fibrotic lesions, wherein pneumothorax did not require thoracic drainage. However, there was no significant difference in the rate of pneumothorax between patients with PPLs within or near fibrotic lesions and those with PPLs distant from fibrotic lesions (16.1% vs. 8.3%, *p* = 0.451).

## 4. Discussion

Our results revealed that the diagnostic yield for PPLs within or near fibrotic lesions was lower than that for PPLs distant from fibrotic lesions. Moreover, the positional relation of PPLs to fibrotic lesions and the probe position on EBUS images were significant factors associated with the diagnostic yield for PPLs in patients with ILD.

Lung cancer with ILD often presents as PPLs within or near fibrotic lesions [2]. We hypothesised that bronchoscopic diagnosis of PPLs within or near fibrotic lesions would be difficult compared to those distant from fibrotic lesions, owing to the difficulty of reaching the lesions in relation to anatomical structures (e.g., the obtuse angle of the bronchial branch) and the difficulty in identifying lesions by fluoroscopy. However, in our study, the rate at which the probe was located within the PPL was almost the same in patients with PPLs within or near fibrotic lesions and those with PPLs distant from fibrotic lesions (51.6% vs. 50%). This result mirrored the fact that the probe reached the PPLs appropriately in both groups with nearly the same frequency. However, when the probe was located within the lesion, PPLs within or near fibrotic lesions had lower diagnostic yield than those distant from fibrotic lesions. A previous report demonstrated that pulmonary infiltrates, like inflammatory cells and fibrotic changes, increased in the side in which lung cancer developed compared to the side that did not [23]. In our study, when the probe was located within the lesion, most cases of malignant lesions unsuccessfully diagnosed by EBUS-GS TBB showed inflammatory cells and fibrotic changes with no malignant cells (Figure 3). Although the target lesions can be accessed accurately under the navigation system and contact with the lesions can be confirmed on EBUS, this finding may reflect the fact that only the surface of PPLs within or near fibrotic lesions are being sampled. To improve the diagnostic yield for PPLs within or near fibrotic lesions, methods of additional conventional transbronchial biopsy after EBUS-GS TBB may be effective by obtaining larger samples [24]. Furthermore, transbronchial needle aspiration through a GS using the PeriView FLEX transbronchial needle aspiration device may improve diagnostic yield by penetrating the lesion and collecting samples [25,26,27].

In previous reports, the probe position on EBUS images was a significant factor affecting the diagnostic yield for PPLs by EBUS-GS TBB [11,13]. Similarly, our study demonstrated that in patients with ILD, the probe position on EBUS images was a significant factor affecting the diagnostic yield of PPLs by EBUS-GS TBB. Moreover, we demonstrated that the positional relation of PPLs to fibrotic lesions was also a significant factor in the successful diagnosis of PPLs in patients with ILD. It has been reported that the diagnostic yield of outer lesions, which are farther from the hilum, is lower than that of inner lesions because it is more difficult to properly guide the biopsy forceps to the lesion [11,28]. However, in patients with ILD, the positional relation of PPLs to fibrotic lesions rather than the lesion classification of the inner or outer lesions was a significant factor affecting the successful diagnostic yield using EBUS-GS TBB.

The rate of pneumothorax with EBUS-GS TBB was reported to be less than 1% [29,30]. Hayama et al. reported that in EBUS-GS procedures which were performed for diagnosing 965 PPLs, there were eight patients (0.8%) with pneumothorax, and three (0.3%) of them required chest tube drainage [29]. In our patients with ILD, the rate of pneumothorax and those that required chest tube drainage were 12.7% and 5.5%, respectively, which were higher than that reported with EBUS-GS TBB in previous studies. We believe that in patients with ILD, the damage to alveolar tissue during biopsy is related to pneumothorax. Furthermore, although we considered that there was no significant difference between the complications observed in patients with PPLs within or near fibrotic lesions and those with PPLs distant from fibrotic lesions because of the small sample size, we found that pneumothorax occurred at a higher frequency in patients with PPLs near or within fibrotic lesions in our study than in that reported in previous studies [29,30]. However, the frequency of pneumothorax and the requirement of drainage with EBUS-GS TBB in patients with PPLs within or near fibrotic lesions was acceptable (16.1% and 9.7%, respectively) compared to that with TTNB, which was reported as 26.7% and 16.9%, respectively [31].

Our study had certain limitations. In the clinical backgrounds between the two groups, age in patients with PPLs within or near fibrotic lesions was significantly lower than that in patients with PPLs distant from fibrotic lesions. Park et al. reported that there was no significant difference in age between lung cancer located inside the fibrotic area and that developed in the nonfibrotic area [18]. In our study, unlike the report, PPLs contained not only lung cancer but also benign lesions. We considered the reasons for a significant difference in age between both groups were the difference of disease groups which we analysed, and bias associated with a retrospective and small cohort study at a single facility. Second, we did not fully evaluate the techniques to overcome the lower diagnostic yield of PPLs within or near fibrotic lesions (e.g., additional conventional transbronchial biopsy). Larger sample sizes and prospective studies are needed for further research.

## 5. Conclusions

In conclusion, the diagnostic yield for PPLs within or near fibrotic lesions was significantly lower than for those distant from fibrotic lesions. Moreover, the positional relation of PPLs to fibrotic lesions and the probe position on the EBUS image were significant factors affecting diagnostic yield.

## Figures and Tables

**Figure 1 cancers-13-05751-f001:**
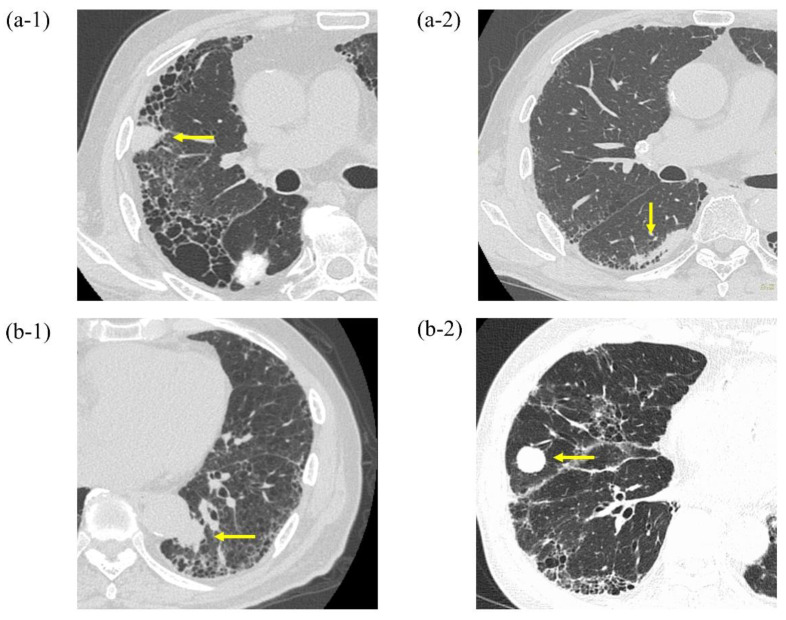
Chest computed tomography showing peripheral pulmonary lesions within or near fibrotic lesions (**a-1**,**a-2**) and those distant from fibrotic lesions (**b-1**,**b-2**).

**Figure 2 cancers-13-05751-f002:**
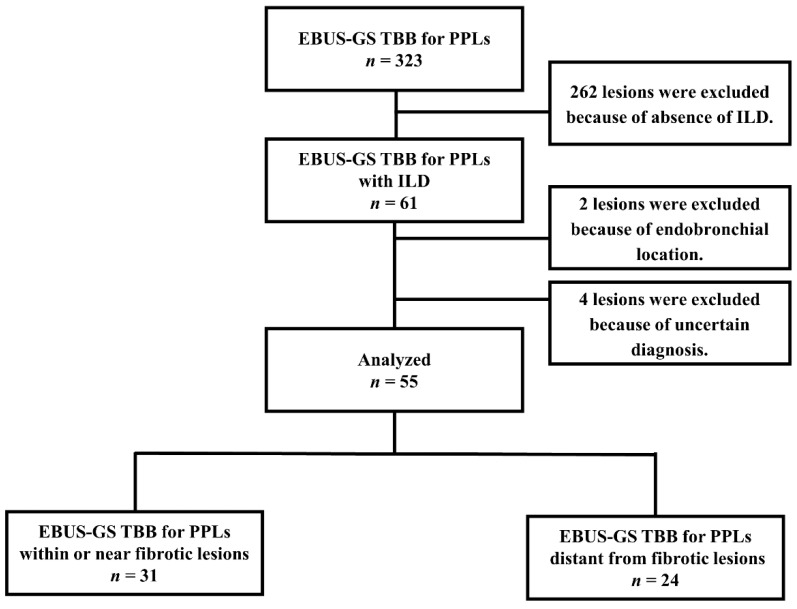
Flow diagram of our study.

**Figure 3 cancers-13-05751-f003:**
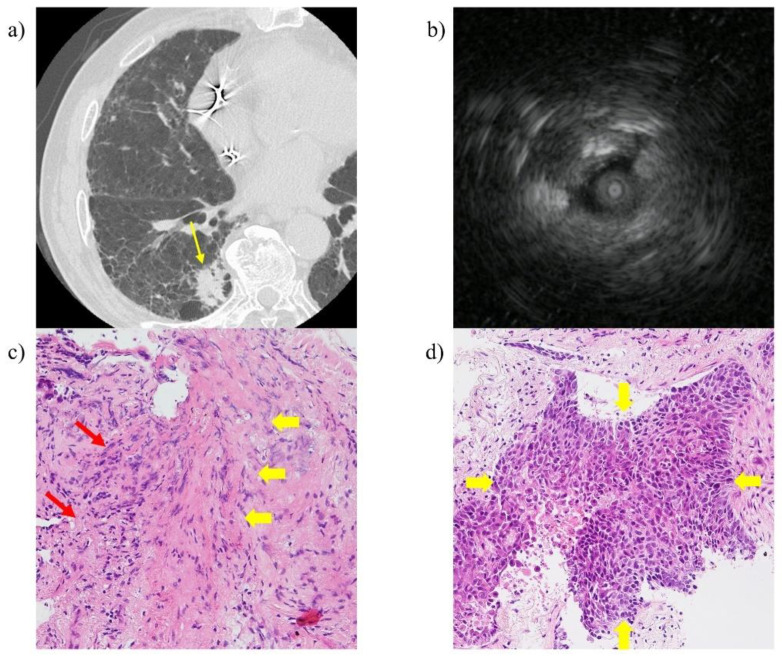
A representative case of a 75-year-old male who was undiagnosed by endobronchial ultrasound with a guide sheath transbronchial biopsy despite collecting samples from a position where the probe is located within the lesion. (**a**) High-resolution computed tomography showed a 28 mm solid nodule (yellow arrow) in the right S6, located near fibrotic lesions. (**b**) The radial endobronchial ultrasound probe was located within the lesion. (**c**) High-power view of the haematoxylin and eosin staining of the specimen obtained by endobronchial ultrasound with a guide sheath transbronchial biopsy showed no malignant cells but inflammatory cells (red arrow) and fibrous thickening (yellow arrow) were present (haematoxylin and eosin staining). (**d**) High-power view of the haematoxylin and eosin staining of the specimen obtained by transthoracic needle biopsy showed malignant cells (yellow arrow).

**Table 1 cancers-13-05751-t001:** Characteristics of the patients and their lesions in the two groups.

Variables	PPLs within or near Fibrotic Lesions *n* = 31	PPLs Distant from Fibrotic Lesions *n* = 24	*p*-Value
Age, years, median (range)	71 (55–84)	75 (58–84)	0.028
Sex, male, *n* (%)	25 (80.6)	17 (70.8)	0.396
Smoking history *n* (%)	23 (74.2)	9 (37.5)	0.006
Classification of ILD			<0.001
IPF, *n* (%)	17 (54.8)	2 (8.3)	
Lesion size, mm, median (range)	23 (9–39)	23 (13–54)	0.925
Lesion lobe, *n* (%)			0.832
Right upper/left upper	10 (32.2)	9 (37.5)	
Right middle/lingula	3 (9.7)	3 (12.5)	
Right lower/left lower	18 (58.1)	12 (50)	
Lesion location, *n* (%)			0.002
Outer	23 (74.2)	8 (33.3)	
Lesion structure, *n* (%)			0.188
Solid	27 (87.1)	24 (100)	
Part-solid	3 (9.7)	0 (0)	
Pure ground-glass	1 (3.2)	0 (0)	
Bronchus sign, *n* (%)			0.122
Positive	25 (80.6)	23 (95.8)	-
FVC, percent predicted, %, median (range)	100 (64–147)	93 (38–130)	0.317
DL_CO_, percent predicted, %, median (range)	71 (36–25)	80 (43–133)	0.302
Visibility on chest X-ray, *n* (%)			0.122
Visible	25 (80.6)	23 (95.8)	
Number of biopsies, median (range)	7.1 (3–14)	6.8 (3–9)	0.642
Malignant lesions, *n* (%)	24 (77.4)	15 (62.5)	0.227

PPLs: peripheral pulmonary lesions; IPF: idiopathic pulmonary fibrosis; FVC: forced vital capacity; DL_CO_: diffusing capacity of the lung for carbon monoxide.

**Table 2 cancers-13-05751-t002:** Diagnostic yields by probe location and disease type.

	PPLs within or near Fibrotic Lesions *n* = 31	PPLs Distant from Fibrotic Lesions *n* = 24	*p*-Value
EBUS image, *n* (%)			
Within	9/16 (56.3)	12/12 (100)	0.023
Adjacent to	2/6 (33.3)	5/6 (83.3)	0.242
Invisible	3/9 (33.3)	3/6 (50)	0.622
Final diagnosis, *n* (%)			
Malignant lesions	10/24 (41.7)	15/15 (100)	<0.001
Benign lesions	4/7 (57.1)	5/9 (55.6)	0.671
Total	14/31 (45.2)	20/24 (83.3)	0.004

Data are shown as numbers of lesions/total lesions (%). PPLs: peripheral pulmonary lesions; EBUS: endobronchial ultrasound.

**Table 3 cancers-13-05751-t003:** Multivariate logistic regression analysis of factors affecting diagnostic yield using EBUS-GS TBB in patients with ILD.

Variables	Reference	OR (95% CI)	*p*-Value
Positional relation to fibrotic lesions PPLs distant from fibrotic lesions (*n* = 24)	PPLs within or near fibrotic lesions (*n* = 31)	7.509 (1.856–30.381)	0.005
Size ≤ 20 mm (*n* = 23)	>20 mm (*n* = 32)	0.985 (0.266–3.642)	0.982
Location inner (*n* = 24)	outer (*n* = 31)	1.239 (0.341–4.509)	0.745
Bronchus sign positive (*n* = 48)	negative (*n* = 7)	0.950 (0.034–26.727)	0.976
EBUS image within (*n* = 28)	others (*n* = 27)	4.172 (1.077–16.167)	0.039

EBUS-GS TBB: endobronchial ultrasonography with a guide sheath transbronchial biopsy; ILD: interstitial lung disease; OR: odds ratio; CI: confidence interval; PPLs: peripheral pulmonary lesions; EBUS: endobronchial ultrasound.

## Data Availability

The data that support the findings of this study are available from the corresponding author upon reasonable request.

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
