# Peer review of "Endobronchial Ultrasonography with a Guide Sheath Transbronchial Biopsy for Diagnosing Peripheral Pulmonary Lesions within or near Fibrotic Lesions in Patients with Interstitial Lung Disease"

_cancers, 2021, doi:10.3390/cancers13225751_

Round 1

Reviewer 1 Report

I would like to congratulate the authors for this elegant and informative manuscript. Herein, I have made some suggestions that, in my opinion, could help to further improve the overall quality of the manuscript.

Abstract

  • The authors may consider defining the cut-off distance for a peripheral pulmonary lesion to be considered near to, as opposed to distant from, a fibrotic lesion.
  • “The diagnostic yield for PPLs-FL was lower than for PPLs-NFL. Moreover, the proximity of PPLs to fibrotic lesions and the probe position were significant factors affecting the yield in patients with ILD.” These sentences are repeated. The authors may consider rephrasing them or omitting them altogether.

Materials and Methods

  • The authors may consider explaining the reasons, if any, of the chosen dates for inclusion of patients in the study (for example, date of introduction of radial EBUS in their department).
  • Again, the authors may consider defining the cut-off distance for a peripheral pulmonary lesion to be considered distant from a fibrotic lesion.
  • The authors may consider explaining why they chose the size of 2 cm as a cut-off value in the multivariate logistic regression analysis (for example, based on a preliminary analysis).

Results

  • The authors may consider omitting Table 4 to avoid duplication of its content, which is presented in the subsection Complications of the “Results”.

Discussion

  • The authors discuss the rate of pneumothorax with various biopsy methods. They may also consider discussing the rate of pneumothorax requiring drainage, as this can be more relevant for patients and clinicians.

Author Response

Abstract

Comment 1: The authors may consider defining the cut-off distance for a peripheral pulmonary lesion to be considered near to, as opposed to distant from, a fibrotic lesion.

Response: We would like to thank Reviewer for the time and effort in reviewing our manuscript and providing comments and suggestions, which have considerably helped us improve our manuscript. We have answered each of your points below and hope that our responses and revisions address all your comments.

In accordance with the previous report, we defined PPLs as within or near fibrotic lesions in our study. PPLs within or near fibrotic lesions met the criterion of the PPLs and the area of fibrotic lesions overlapping each other. We did not define the cut-off distance in our study because of this predetermined criterion and the previous report. Based on your comment, we have added this content to the revised manuscript accordingly in lines 32-34.

Comment 2: The diagnostic yield for PPLs-FL was lower than for PPLs-NFL. Moreover, the proximity of PPLs to fibrotic lesions and the probe position were significant factors affecting the yield in patients with ILD.” These sentences are repeated. The authors may consider rephrasing them or omitting them altogether.

Response: As per your valuable suggestion, we have rephrased the sentences with the support of native English reviewers accordingly in lines 38-39 of the revised manuscript.

Materials and Methods

Comment 1: The authors may consider explaining the reasons, if any, of the chosen dates for inclusion of patients in the study (for example, date of introduction of radial EBUS in their department).

Response: We introduced radial EBUS in our department on 1 November 2014. Since then, we have retrospectively analysed patients who underwent EBUS-GS TBB for diagnosing PPLs for nearly two years. Therefore, we have added an explanation regarding the dates chosen for the inclusion of patients in our study in line 73 of the revised manuscript.

Comment 2: Again, the authors may consider defining the cut-off distance for a peripheral pulmonary lesion to be considered distant from a fibrotic lesion.

Response: In accordance with the previous report, we defined PPLs as distant from fibrotic lesions in our study. PPLs distant from fibrotic lesions met the criterion of the PPLs and the area of fibrotic lesions not overlapping each other. We did not define the cut-off distance in our study because of this predetermined criterion and the previous report. To address your comments, we have added this content to the revised manuscript accordingly in lines 83-85.

Comment 3: The authors may consider explaining why they chose the size of 2 cm as a cut-off value in the multivariate logistic regression analysis (for example, based on a preliminary analysis).

Response: In a meta-analysis, Wang Memoli et al. evaluated the effect of size on the diagnostic yield using radial EBUS for diagnosing PPLs. They reported that 2 cm was frequently used as the size criterion in most previous reports. Therefore, we selected 2 cm as the cut-off value for size in the multivariate logistic regression analysis. As per your suggestion, we have added the explanation for selecting 2 cm as a cut-off value for size in the multivariate logistic regression analysis in lines 143-145 of the revised manuscript .

Results

Comment 1: The authors may consider omitting Table 4 to avoid duplication of its content, which is presented in the subsection Complications of the “Results”.

Response: As suggested by the reviewer, we have omitted Table 4 from the manuscript to avoid duplication of content.

Discussion

Comment 1: The authors discuss the rate of pneumothorax with various biopsy methods. They may also consider discussing the rate of pneumothorax requiring drainage, as this can be more relevant for patients and clinicians.

Response: Based on your suggestion, we have discussed the rate of pneumothorax that required drainage associated with biopsy using EBUS-GS or TTNA based on previous reports in lines 250-264 of the revised manuscript. Hayama et al. reported that in EBUS-GS procedures which were performed for diagnosing 965 PPLs, there were 8 patients (0.8%) with pneumothorax, and 3 (0.3%) of them required chest tube drainage. In our patients with ILD, the rate of pneumothorax and those that required chest tube drainage were 12.7% and 5.5%, respectively, which were higher than that reported with EBUS-GS TBB in previous studies. However, the frequency of pneumothorax and the requirement of drainage with EBUS-GS TBB in patients with PPLs within or near fibrotic lesions was acceptable (16.1% and 9.7%, respectively) compared to that with TTNB, which was reported as 26.7% and 16.9%, respectively.

Reviewer 2 Report

Ito and colleagues report a retrospective study to evaluate the utility and safety of endobronchial ultrasonography with a guide sheath transbronchial biopsy for diagnosing peripheral pulmonary lesions within or near fibrotic lesions in patients with interstitial lung disease. I have some comments.

  • In figure 2, the authors state that lesions with uncertain diagnosis were excluded from this study. However, only 14 out of 31 lesions of PPLs-FL have been diagnosed in table 2. In the 17 lesions where no diagnosis was made, what additional procedures were performed and what was the final diagnosis? In addition, what was the positive diagnosis rate and safety of the additional procedures performed? I think it is important to have that information when considering whether to implement EBUS-GS TBB for PPLs-FL.
  • In table 3, multivariate analysis is performed. The number of factors in the analysis is 5, which seems a bit large for a sample size of 55 and 34 events. Have you checked with a biostatistician about the validity of the analysis?
  • Were there any cases where rapid on-site cytologic evaluation was performed? If they are mixed, I think that would be a major bias.

Author Response

Comment 1: In figure 2, the authors state that lesions with uncertain diagnosis were excluded from this study. However, only 14 out of 31 lesions of PPLs-FL have been diagnosed in table 2. In the 17 lesions where no diagnosis was made, what additional procedures were performed and what was the final diagnosis? In addition, what was the positive diagnosis rate and safety of the additional procedures performed? I think it is important to have that information when considering whether to implement EBUS-GS TBB for PPLs-FL.

Response: We would like to thank Reviewer for the time and effort in reviewing our manuscript and providing comments and suggestions, which have considerably helped us improve our manuscript. We have answered each of your points below and hope that our responses and revisions address all your comments.

Of the 17 undiagnosed lesions, 10 were diagnosed using TTNA, whereas four were diagnosed using SLB. Based on your comment, we have added the final pathological diagnosis of the remaining 14 lesions in the revised manuscript. We apologise for any confusion. Of the 17 lesions, three lesions were assessed via follows-ups with CT, and all the three lesions regressed spontaneously six months after bronchoscopy. Therefore, these lesions were diagnosed as inflammatory lesions.

Furthermore, as per your suggestion, we have added the positive diagnosis rate [TTNA: 10/10 (100%) and SLB: 4/4 (100%)] and the rate of complication [TTNA: 3/10 (30%) and SLB: 0/4 (0%)] of the additional procedures to the revised manuscript in lines 173-185.

Comment 2: In table 3, multivariate analysis is performed. The number of factors in the analysis is 5, which seems a bit large for a sample size of 55 and 34 events. Have you checked with a biostatistician about the validity of the analysis?

Response: The statistical analysis in our study was checked by a biostatistician (F.K). We considered that one-fifth to one-tenth in the less size of event number was able to be forced into the logistic regression model. Furthermore, considering our study was an exploratory study, we thought a maximum of 5 factors to be acceptable.

Comment 3: Were there any cases where rapid on-site cytologic evaluation was performed? If they are mixed, I think that would be a major bias.

Response: In our study, we did not use rapid on-site evaluation during EBUS-GS TBB. To address your comment, we have added an explanation regarding this in lines 105-106 of the revised manuscript accordingly.

Reviewer 3 Report

In the manuscript „Endobronchial ultrasonography with a guide sheath transbronchial biopsy for diagnosing peripheral pulmonary lesions within or near fibrotic lesions in patients with interstitial lung disease” by Ito T. et al the authors assessed the success rate of EBUS-GS biopsies in the patients with peripheral lung lesions depending on the localization of the lesion according to the fibrotic areas. They have demonstrated that the diagnostic yield for lesions found outside of the fibrotic areas is higher than for the ones in fibrotic areas. This issue is very important and in the current literature not (adequately) addressed.

Overall I compliment the authors on their work. However, prior to acceptance there are several points authors should address:

  1. In lines 47-49, the authors are shortly mentioning issues with treating malignant lesions, and only at the end do they conclude that the histologic diagnosis is necessary. They should rewrite this part since not all peripheral lung lesions are malignant, so first, it is important to make diagnosis, and then to mention issues with surgery/chemo/radiotherapy.
  2. In line 55, please change compared with…to compared to
  3. In Patient enrolment, it should be clearly stated that the group of interest are ILD patients with EBUS-GS, and then describe how they were selected.
  4. In lines 110-112, the authors defined successful diagnosis when it was confirmed as malignant. In my opinion, if the lesion was obtained and diagnosed as benign, it was also a successful biopsy. Please rewrite this part to make it more clear.
  5. In lines 132-136, I would suggest rewriting: Among the 323 PPLs evaluated by EBUS-GS TBB, 61 lesions were identified in patients with ILD. After excluding 2 endobronchial lesions and 4 lesions because of uncertain diagnosis, 55 lesions were included in the analyses. Among these, 31 PPLs within or near fibrotic lesions and 24 distant from fibrotic lesions (Figure 2). The characteristics of the patients in the two groups are presented in Table 1.
  6. In Figure 2, please change: 2 lesions were excluded because of endobronchial lesions …to…2 lesions were excluded because of endobronchial location
  7. Also in line 135/136 change The characteristics of the patients in the two groups are presented in Table 1…..to…. The characteristics of the patients and their lesions in the two groups are presented in Table 1.
  8. Consider also adaptation of the Table 1 title: Clinical backgrounds between PPLs within or near fibrotic lesions and those dittant from fibrotic lesions; In the Table 1 you are not presenting only Clinical data, but also data about lesion radiologically and histologically.
  9. From line 169, in Complications, please when writing about patients do not use PPL- because now it sounds like PPL had pneumothorax, and actually, patients did. The same goes for lines 229-231.
  10. In lines 188-190, I do not quite understand what was intended to say? “However, in our study, the rate in which the probe was located within the PPL resulted in almost the same consequence between PPLs within or near fibrotic lesions and those distant from fibrotic lesions.” Did the authors want to state that the rate was the same? Then please state this so, and do not mention consequence in this.
  11. In lines 203-204 I would suggest deleting this: effective for improving the lower diagnostic yield compared with EBUS-GS TBB alone….because it is repeating the first part of this sentence
  12. In line 211 please change emphasized with some other verb (maybe showed, demonstrated, found…)
  13. In lines 222-225, rewriting is needed. I think that the authors wanted to state that they have found pneumothorax in higher frequency in patients with PPL near or within fibrotic lesions. As it is now written it is not very clear.
  14. In the description of Figure 3, please change needed to needle in line 250

Author Response

Comment 1: In lines 47-49, the authors are shortly mentioning issues with treating malignant lesions, and only at the end do they conclude that the histologic diagnosis is necessary. They should rewrite this part since not all peripheral lung lesions are malignant, so first, it is important to make diagnosis, and then to mention issues with surgery/chemo/radiotherapy.

Response: We would like to thank Reviewer for the time and effort in reviewing our manuscript and providing comments and suggestions, which have considerably helped us improve our manuscript. We have answered each of your points below and hope that our responses and revisions address all your comments.

As suggested by the reviewer, we have stated the importance of obtaining a correct histological diagnosis by sampling lesions, given that all lesions occurring in patients with ILD were not always malignant. For clarity, we have mentioned the issues associated with surgery/chemo/radiotherapy in patients with ILD after these sentences in lines 47-52 of the revised manuscript.

Comment 2: In line 55, please change compared with…to compared to.

Response: As per your suggestion, we have replaced ‘compared with’ with ‘compared to’ in lines 57-58 of the revised manuscript.

Comment 3: In Patient enrolment, it should be clearly stated that the group of interest are ILD patients with EBUS-GS, and then describe how they were selected.

Response: As suggested by the reviewer, we have stated that the group of interest are patients with ILD subjected to EBUS-GS. In addition, we have added a description of how they were selected to the revised manuscript in lines 78-82.

Comment 4: In lines 110-112, the authors defined successful diagnosis when it was confirmed as malignant. In my opinion, if the lesion was obtained and diagnosed as benign, it was also a successful biopsy. Please rewrite this part to make it more clear.

Response: For clarity, we have defined the criteria of successful diagnosis of benign lesions as the confirmation of specific benign findings using EBUS-GS in lines 121-131 of the revised manuscript.

Comment 5: In lines 132-136, I would suggest rewriting: Among the 323 PPLs evaluated by EBUS-GS TBB, 61 lesions were identified in patients with ILD. After excluding 2 endobronchial lesions and 4 lesions because of uncertain diagnosis, 55 lesions were included in the analyses. Among these, 31 PPLs within or near fibrotic lesions and 24 distant from fibrotic lesions (Figure 2). The characteristics of the patients in the two groups are presented in Table 1.

Response: As per your suggestion, we have rewritten these sentences with the support of native English reviewers accordingly in lines 150-155 of the revised manuscript.

Comment 6: In Figure 2, please change: 2 lesions were excluded because of endobronchial lesions …to…2 lesions were excluded because of endobronchial location.

Response: We have replaced ‘2 lesions were excluded because of endobronchial lesions’ with ‘2 lesions were excluded because of endobronchial location’ in Figure 2.

Comment 7: Also in line 135/136 change The characteristics of the patients in the two groups are presented in Table 1…..to…. The characteristics of the patients and their lesions in the two groups are presented in Table 1.

Response: As per your suggestion, we have replaced ‘The characteristics of the patients in the two groups are presented in Table 1’ with ‘The characteristics of the patients and their lesions in the two groups are presented in Table 1’ in lines 154-155 of the revised manuscript.

Comment 8: Consider also adaptation of the Table 1 title: Clinical backgrounds between PPLs within or near fibrotic lesions and those distant from fibrotic lesions; In the Table 1 you are not presenting only Clinical data, but also data about lesion radiologically and histologically.

Response: We have changed the title of Table 1 accordingly to ‘Characteristics of the patients and their lesions in the two groups’.

Comment 9: From line 169, in Complications, please when writing about patients do not use PPL- because now it sounds like PPL had pneumothorax, and actually, patients did. The same goes for lines 229-231.

Response: As suggested by the reviewer, we have rewritten these sentences with the support of native English reviewers in lines 200 and 265-267.

Comment 10: In lines 188-190, I do not quite understand what was intended to say? “However, in our study, the rate in which the probe was located within the PPL resulted in almost the same consequence between PPLs within or near fibrotic lesions and those distant from fibrotic lesions.” Did the authors want to state that the rate was the same? Then please state this so, and do not mention consequence in this.

Response: We apologise for using ambiguous expressions. We have already stated that the rate at which the probe was located within the PPL was almost the same in the patients with PPLs within or near fibrotic lesions and those with PPLs distant from fibrotic lesions (51.6% vs. 50%) in lines 148-150 of the original manuscript. EBUS images helped us to judge whether the probe reached the lesion appropriately. It is crucial to perform the biopsy in a position where the probe is located within the lesion rather than in a position where the probe is located adjacent to or outside the lesion. As per your suggestion, we have rewritten the relevant text in the Results section with the support of native English reviewers.

Comment 11: In lines 203-204 I would suggest deleting this: effective for improving the lower diagnostic yield compared with EBUS-GS TBB alone….because it is repeating the first part of this sentence.

Response: As suggested by the reviewer, we have removed the expression ‘for improving the lower diagnostic yield compared with EBUS-GS TBB alone’ from the manuscript.

Comment 12: In line 211 please change emphasized with some other verb (maybe showed, demonstrated, found…)

Response: As suggested by the reviewer, we have replaced ‘emphasized’ with ‘demonstrated’ in line 242 of the revised manuscript.

Comment 13: In lines 222-225, rewriting is needed. I think that the authors wanted to state that they have found pneumothorax in higher frequency in patients with PPL near or within fibrotic lesions. As it is now written it is not very clear.

Response: As per your suggestion, we have rewritten these sentences concisely and appropriately with the support of native English reviewers in lines 256-260 of the revised manuscript. We hope that the text is now clear.

Comment 14: In the description of Figure 3, please change needed to needle in line 250.

Response: As suggested by the reviewer, we have replaced ‘needed’ with ‘needle’ in line 287 of the revised manuscript.

Round 2

Reviewer 2 Report

In the current manuscript, the points commented on in the previous version have been appropriately corrected. But isn't TTNA in lines 198 to 210 referring to TTNB? Please just check that point. Other than that, I think everything is fine.